# UɴɪDEC : Unified Dual Encoder and Classifier Training for Extreme Multi-Label Classification

## Abstract

Extreme Multi-label Classification (XMC) involves predicting a subset of relevant labels from an extremely large label space, given an input query and labels with textual features. Models developed for this problem have conventionally made use of dual encoder (DE) to embed the queries and label texts and one-vs-all (OvA) classifiers to rerank the shortlisted labels by the DE. While such methods have shown empirical success, a major drawback is their computational cost, often requiring upto 16 GPUs to train on the largest public dataset. Such a high cost is a consequence of calculating the loss over the entire label space. While shortlisting strategies have been proposed for classifiers, we aim to study such methods for the DE framework. In this work, we develop UniDEC, a loss-independent, end-to-end trainable framework which trains the DE and classifier together in a unified manner with a multi-class loss, while reducing the computational cost by $4-16\times$. This is done via the proposed *pick-some-label (PSL)* reduction, which aims to compute the loss on only a subset of positive and negative labels. These labels are carefully chosen in-batch so as to maximise their supervisory signals. Not only does the proposed framework achieve state-of-the-art results on datasets with labels in the order of millions, it is also computationally and resource efficient in achieving this performance on a single GPU. Code is provided with the submission and will be open-sourced upon acceptance.

### ACM Reference Format:
Anonymous Author(s). 2018. UɴɪDEC : Unified Dual Encoder and Classifier Training for Extreme Multi-Label Classification. In *Proceedings of Make sure to enter the correct conference title from your rights confirmation emai (Conference acronym 'XX)*. ACM, New York, NY, USA, 11 pages. https://doi.org/XXXXXXX.XXXXXXX

## 1 Introduction

Extreme Multi-label Classification (XMC) is described as the task of identifying i.e. retrieving a subset, comprising of one or more labels, that are most relevant to the given data point from an extremely large label space, potentially consisting of millions of possible choices. Over time, XMC has increasingly found its relevance for solving multiple real world use cases. Typically, *long-text XMC* approaches are leveraged for the tasks of document tagging and product recommendation and *short-text XMC* approaches target tasks such as query-ad keyword matching and related query recommendation. Notably, in the real world manifestations of these

rt

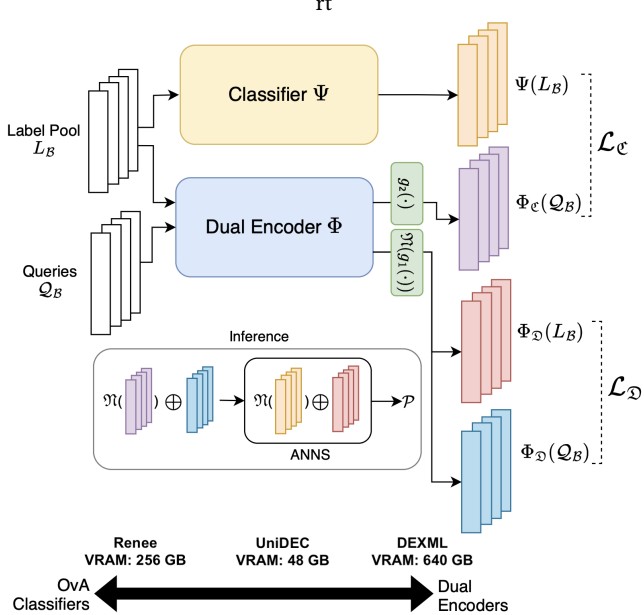

**Figure 1: The architecture for the UɴɪDEC framework, denoting the the classifiers and DE trained in parallel, along with the loss functions used. The inference pipeline is shown in the rectangular box.**

use cases, the distribution of instances among labels exhibits a fit to Zipf's law [1]. This implies, the vast label space ($L \approx 10^6$) is skewed and is characterized by the existence of *head*, *torso* and *tail* labels [33]. For example, in query-ad keyword matching for search engines like Bing, Google etc. head keywords are often exact match or related phrase extensions of popularly searched queries while tail keywords often target specific niche queries. Typically, we can characterize head, torso and tail keywords as having $> 100$, 10 - 100, and 1 - 10 annotations, respectively.

Typically, per-label classifiers are employed for solving the XMC task. A naive strategy to train classifiers for XMC involves calculating the loss over the entire label set. This method has seen empirical success, particularly in earlier works like Dɪsᴍᴇᴄ [2], which learns one-vs-all classifiers by parallelisation across multiple CPU cores. With the adoption of deep encoders, various works moved to training on GPUs, which due to VRAM constraints, led to the use of tree-based shortlisting strategies [4, 7, 18, 19] to train classifiers on the hardest negatives. This reduced the computational complexity from $O(L)$ to $O(\log L)$. While leveraging a label shortlist made XMC training more feasible via modular training [6, 8] or joint training using a meta-classifier [15, 18, 19] , it still left out scope for empirical improvements. Consequently, Rᴇɴᴇᴇ [13] demonstrated

the extreme case for methods which employ classifiers with deep encoders by writing custom CUDA kernels to scale classifier training over the entire label space. This, however, leads to a GPU VRAM usage of 256GB (Figure 1) for training a DISTILBERT model on LF-AmazonTitles-1.3M, the largest XMC dataset. Notably, what remains common across XMC classifier-training algorithms is the advocacy of OvA reduction for the multi-label problem [5]. Theoretically, the alternative, pick-all-labels (PAL), should lead to better optimization over OvA, since it promotes "competition" amongst labels [22]. However, PAL has neither been well-studied nor successfully leveraged to train classifiers in XMC since such losses require considering all labels, which is prohibitively expensive.

A parallel line of research involves leveraging dual encoders (DE) for XMC. While DE models are a popular choice for dense retrieval (DR) and open-domain question answering (ODQA) tasks, these are predominantly *few* and *zero-shot* scenarios. In contrast, XMC covers a broader range of scenarios (see Appendix B). Consequently, modelling the XMC task as a retrieval problem is tantamount to training a DE simultaneously on *many*, *few* and *one-shot* scenarios. While DE trained with triplet loss was thought to be insufficient for XMC, and thus augmented with per-label classifiers to enhance performance [6, 11], a recent work DEXML [11] proved the sufficiency of the DE framework for XMC by proposing a new multi-class loss function *Decoupled Softmax*, which computed the loss over the entire label space. This, however is very computationally expensive, as DEXML requires 640GB VRAM to train on LF-AmazonTitles-1.3M.

At face value, PAL reduction of multi-label problems for DE training should be made tractable by optimizing over in-batch labels, however in practice, it does not scale to larger datasets due to the higher number of positives per label. For instance, for LF-AmazonTitles-1.3M a batch consisting of 1,000 queries will need an inordinately large label pool of size $\sim$ 22.2K (considering in-batch negatives) to effectively train a DE with the PAL loss. Alternatively, the stochastic implementation of PAL in the form of pick-one-label (POL) reduction used by DEXML, either convergences slowly [11] or fails to reach SOTA performance.

In order to enable efficient training, in this work, we propose ***"pick-some-labels"*** *(PSL)* relaxation of the PAL reduction for the multi-label classification problem which enables scaling to large datasets ($\sim 10^6$ labels). Here, instead of trying to include all the positive labels for instances in a batch, we propose to randomly sample at max $\beta$ positive labels per instance. To the best of our knowledge, we are the first work to study the effect of multi-class losses for training classifiers at an extreme scale. Further, we aim to develop an end-to-end trainable loss-independent framework, UNIDEC - **Uni**fied **D**ual **E**ncoder and **C**lassifier, for XMC that leverages the multi-positive nature of the XMC task to create highly informative in-batch labels to train the DE, and be used as a shortlist for the classifier. As shown in Figure 1, UNIDEC, in a single pass, performs an update step over the combined loss computed over two heads: (i) between DE head's query and sampled label-text embeddings, (ii) between classifier (CLF) head's query embeddings and classifier weights corresponding to sampled labels. By unifying the two compute-heavy ends of the XMC spectrum in such a way, UNIDEC is able to significantly reduce the training computational cost down to a **single 48GB GPU**, even for the largest dataset with 1.3M labels. End-to-end training offers multiple benefits as it (i)

helps us do away with a meta-classifier and modular training, (ii) dynamically provides progressively harder negatives with lower GPU VRAM consumption, which has been shown to outperform static negative mining [15, 18, 19] (iii) additionally, with an Approximate Nearest Neighbour Search (ANNS), it can explicitly mine hard negative labels added to the in-batch negatives. While UNIDEC is a loss independent framework (see Table 3), the focus of this work also includes studying the use of multi-class losses for training multi-label classifiers at an extreme scale via the proposed *PSL* reduction. To this end, we benchmark UNIDEC on 6 public datasets, forwarding the state-of-the-art in each, and a proprietary dataset containing 450M labels. Finally, we also experimentally show how OvA losses like BCE can be applied in tandem with multi-class losses for classifier training.

## 2 Related Works & Preliminaries

For training, we have a multi-label dataset $\mathcal{D} = \{\{\mathbf{x}_i, \mathcal{P}_i\}_{i=1}^N, \{\mathbf{z}_l\}_{l=1}^L\}$ comprising of $N$ data points and $L$ labels. Each $\mathbf{x}_i$ is associated with a small ground truth label set $\mathcal{P}_i \subset [L]$ out of $L \sim 10^6$ possible labels. Further, $\mathbf{x}_i, \mathbf{z}_l \in \mathcal{X}$ denote the textual descriptions of the data point $i$ and the label $l$ respectively, which, in this setting, derive from the same vocabulary universe $\mathcal{V}$ [5]. The goal is to learn a parameterized function $f$ which maps each instance $\mathbf{x}_i$ to the vector of its true labels $\mathbf{y}_i \in [0, 1]^L$ where $\mathbf{y}_{i,l} = 1 \Leftrightarrow l \in \mathcal{P}_i$.

***Dual Encoder***. A DE consists of the query encoder $\Phi_q$, and a label encoder $\Phi_l$. Conventionally, the parameters for $\Phi_q$ and $\Phi_l$ are shared, and thus we will simply represent it as $\Phi$ [6, 11, 16, 28]. The mapping $\Phi(.)$ projects the instance $\mathbf{x}_i$ and label-text $\mathbf{z}_l$ into a shared $d$-dimensional unit hypersphere $\mathcal{S}^{d-1}$. For each instance $\mathbf{x}_i$, its similarity with label $\mathbf{z}_l$ is computed via an inner product i.e., $s_{i,l} = \langle \Phi(\mathbf{x}_i), \Phi(\mathbf{z}_l) \rangle$ to produce a ranked list of top-K labels.

Training two-tower algorithms for XMC at scale is made possible by recursively splitting (say, via a hierarchical clustering strategy) instance encoder embeddings $\{\Phi(\mathbf{x}_i)\}_{i=1}^N$ into disjoint clusters $\mathfrak{B}$ [6], where each cluster represents a training batch $\mathcal{B}$. Each batch $\mathcal{B} = \{Q_{\mathcal{B}}, L_{\mathcal{B}}\}$ is characterised by a set of instance indices $Q_{\mathcal{B}} = \{i \mid i \subset [N]\}$, $s.t.$ $|Q_{\mathcal{B}}| = N/|\mathfrak{B}|$, and the corresponding collated set of (typically one per instance) sampled positive labels $p \in \mathcal{P}_i$, defined as $L_{\mathcal{B}} = \{p \mid p \in \mathcal{P}_i \text{ and } i \in Q_{\mathcal{B}}\}$. As per the in-batch negative sampling strategy common across existing works [6, 11, 16, 28], the negative label pool then is made up of the positive labels sampled for other instances in the batch i.e. $\mathcal{N}_i = L_{\mathcal{B}} - \mathcal{P}_i$. As compared to random batching, [6] posit that the batches created from instance-clustering are *negative-mining aware* i.e. for every instance, the sampled positives of the other instances in the batch serve as the set of appropriate "hard" negatives.

An additional effect of this is the accumulation of multiple in-batch positives for most queries (see Figure 2a). This makes the direct application of commonly used multi-class loss - InfoNCE loss - infeasible for training DE. Hence XMC methods find it suitable to replace InfoNCE loss with a triplet loss [6, 8] or probabilistic contrastive loss [5], as it can be potentially applied over multiple positives and hard negatives (equation 1 in [6]). While this would seem favourable, these approaches still fail to leverage the additional positive signals owing to multiple positives in the batch as

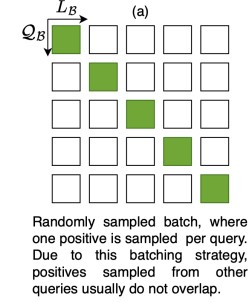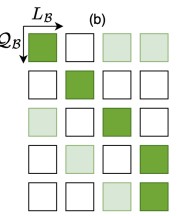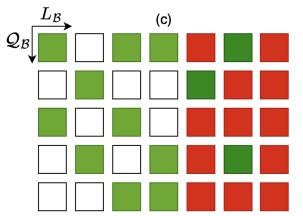

Randomly sampled batch, where one positive is sampled per query. Due to this batching strategy, positives sampled from other queries usually do not overlap.

Negative-mining aware sampled batch, where positives of multiple queries overlap. These additional positives are typically ignored while calculating the loss.

The same batch, when trained with UniDEC. The additional positives are also incorporated in the loss function. The explicitly added hardest negatives included in the batch. The mined hard negatives when collated, overlap and may be positives for other queries.

$\hat{\mathcal{P}}_i = \{$ ▇ $\}$        $\mathcal{P}_i^{\mathcal{B}} = \{$ ▇ ▢ ▇ $\}$        $\hat{\mathcal{H}}_i = \{$ ▇ ▇ $\}$

(a)

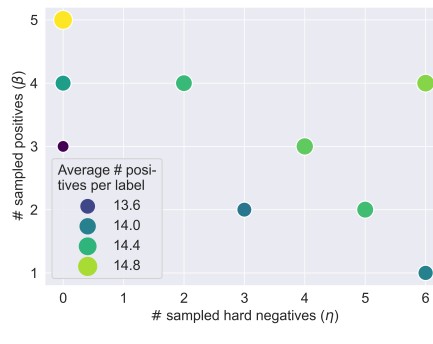

(b)

Figure 2: (a) Visualizing UniDEC's batching strategy. Such a framework naturally leads to higher number of positives per query, enabling us to scale without increasing the batch size significantly. (b) Scatter plot showing the average number of positive labels per query, when we sample $\beta$ positives and $\eta$ hard negatives in the batch. Note that, even with $\beta = 3$ and $\eta = 0$, $\mathbf{avg}(|P|) = 13.6$.

they calculate loss over only a single sampled positive i.e. employing POL reduction instead of PAL reduction.

**Classifiers in XMC.** The traditional XMC set-up considers labels as featureless integer identifiers which replace the encoder representation of labels $\Phi(z_l)$ with learnable classifier embeddings $\Psi_{l=1}^L \in \mathbb{R}^{L \times d}$ [4, 37, 38]. The relevance of a label $l$ to an instance is scored using an inner product, $s_{i,l} = \langle \Phi(\mathbf{x}_i), \Psi_l \rangle$ to select the $k$ highest-scoring labels. Under the conventional OvA paradigm, each label is independently treated with a binary loss function $\ell_{BC}$ applied to each entry in the score vector. It can be expressed as,

$$\mathcal{L}_{\text{OVA}} = \sum_{l=1}^{L} \{y_l \cdot \ell_{BC}(1, s_{i,l}) + (1 - y_l) \cdot \ell_{BC}(0, s_{i,l})\}$$

## 3 Method: UniDEC

In this work, we propose a novel multi-task learning framework which, in an end-to-end manner, trains both - a dual encoder and extreme classifiers - in parallel. The framework eliminates the need of a meta classifier for a dynamic in-batch shortlist. Further, it provides the encoder with the capability to explicitly mine hard-negatives, obtained by querying an ANNS, created over $\{\Phi(\mathbf{z}_l)\}_{l=1}^L$, which is refreshed every $\varepsilon$ epochs.

The DE head is denoted by $\Phi_{\mathfrak{D}}(\cdot) = \mathfrak{N}(g_1(\Phi(\cdot)))$ and the classifier head by $\Phi_{\mathfrak{C}}(\cdot) = g_2(\Phi(\cdot))$, where $\mathfrak{N}$ represents the L2 normalization operator and $g_1(\cdot)$ and $g_2(\cdot)$ represent separate nonlinear projections. Unlike DE, and as is standard practice for OvA classifiers, we train them without additional normalization [5, 7].

### 3.1 *Pick-some-Labels* Reduction

$\mathcal{L}_{\text{PAL-N}}$ [22] is is formulated as :

$$\mathcal{L}_{\text{PAL-N}}(\Phi_1(\mathbf{x}), \Phi_2(z_l)) = \frac{1}{\sum_{j=1}^L y_j} \sum_{l=1}^L y_l \cdot \ell_{MC}(1, \langle \Phi_1(\mathbf{x}), \Phi_2(z_l) \rangle)$$

Since it computes the loss over the entire label space, it is computationally intractable for XMC scenarios. To reduce the computational costs associated with this reduction, we propose a relaxation

by computing loss over some labels in batch $\mathcal{B} = \{Q_{\mathcal{B}}, L_{\mathcal{B}}\}$, which we call *pick-some-labels (PSL)*.

$$\mathcal{L}_{PSL}(\Phi_1(\mathbf{x}), \Phi_2(z_l) \mid \mathcal{B}, \mathcal{P}^{\mathcal{B}}) = \sum_{i \in Q_{\mathcal{B}}} \frac{-1}{|\mathcal{P}_i^{\mathcal{B}}|} \sum_{p \in \mathcal{P}_i^{\mathcal{B}}} \ell_{MC}(1, \langle \Phi_1(\mathbf{x}), \Phi_2(z_p) \rangle)$$

where $\Phi_1$ and $\Phi_2$ are encoding networks. Any multi-class loss [1] can be used in place of $\ell_{MC}$. By varying $\Phi_1$ and $\Phi_2$, we get a generic loss function for training classifier as well as DE. This approximation enables employing PAL-N over a minibatch $Q_{\mathcal{B}}$ by sampling a subset of positive labels $\hat{\mathcal{P}}_i \subseteq \mathcal{P}_i$ s.t. $|\hat{\mathcal{P}}_i| \leq \beta$. Typical value for $\beta$ can be found in Figure 2b. The collated label pool, considering in-batch negative mining, is defined as $L_{\mathcal{B}} = \{\bigcup_{i \in Q_{\mathcal{B}}} \hat{\mathcal{P}}_i\}$. Here, $\mathcal{P}_i^{\mathcal{B}} = \{\mathcal{P}_i \cap L_{\mathcal{B}}\}$ denotes all the in-batch positives for an instance $\mathbf{x}_i$, i.e., the green and pale green in Figure 2.

### 3.2 Dual Encoder Training with *Pick-some-Labels*

The PSL loss to train a DE is formulated as,

$$\mathcal{L}_{\mathfrak{D}, q2l} = \mathcal{L}_{PSL}(\Phi_{\mathfrak{D}}(\mathbf{x}), \Phi_{\mathfrak{D}}(z_l) \mid \mathcal{B}, \mathcal{P}^{\mathcal{B}})$$

More specifically, we perform k-means clustering on the queries such that similar queries are clustered into the same batch $\mathfrak{B}$, leading to both positive and negative-aware batching [6]. Thus, $\mathcal{P}_i^{\mathcal{B}}$ consists not only of the sampled positives $\hat{\mathcal{P}}_i$ but also those non-sampled positives that exist in the batch as sampled positives of other instances i.e. $\mathcal{P}_i^{\mathcal{B}} = \hat{\mathcal{P}}_i \cup \{\bigcup_{j \in \{Q_{\mathcal{B}} - \{i\}\}} \hat{\mathcal{P}}_j \cap \mathcal{P}_i\}$. We find the cardinality of the second term to be non-zero for most instances having $|\mathcal{P}_i| > \beta$ due to a high overlap of sampled positive labels in query-clustered batches, leading to a more optimal batch size. Thus, although we sample $|\hat{\mathcal{P}}_i| \leq \beta \ \forall \ i \in Q_{\mathcal{B}}, \ \exists \ i \in Q_{\mathcal{B}} \ s.t. \ |\mathcal{P}_i^{\mathcal{B}}| \geq \beta$. As per our observations, $\mathcal{P}_i^{\mathcal{B}} = \mathcal{P}_i$ for most tail and torso queries. For e.g., even if $\beta = 1$ for LF-AmazonTitles-1.3M, for $|Q_{\mathcal{B}}| = 10^3$, $\text{Avg}(|\hat{\mathcal{P}}_i|) = [12, 14]$. Thus, it makes *PSL* reduction

---
[1] While binary class loss functions can also be used, in this work, our focus is to study multi-class losses

same as PAL for torso and tail labels and only taking form of *PSL* for head queries.

***Dynamic ANNS Hard-Negative Mining.*** While the above strategy leads to collation of hard negatives in a batch, it might not mine hardest-to-classify negatives [6]. We explicitly add them by querying an ANNS created over $\{\Phi_{\mathfrak{D}}(\mathbf{z}_l)\}_{l=1}^{L}$ for all $\{\Phi_{\mathfrak{D}}(\mathbf{x}_i)\}_{i=1}^{N}$. More specifically, for each instance, we create a list of hard negatives $\mathcal{H}_i = \text{top}_k(\text{ANNS}(\Phi_{\mathfrak{D}}(\mathbf{x}_i)|_{i=1}^{N}, \Phi_{\mathfrak{D}}(\mathbf{z}_l)|_{l=1}^{L})) \; s.t. \; \mathcal{H}_i \cap \mathcal{P}_i = \phi$ (denoted by red in Figure 2). Every iteration, we uniformly sample a $\eta$-sized hard-negative label subset $\hat{\mathcal{H}}_i \subset \mathcal{H}_i$ alongside $\hat{\mathcal{P}}_i \; \forall \; \mathbf{x}_i \in Q_{\mathcal{B}}$. More formally, the new batch label pool can be denoted as $L_{\mathcal{B}} = \{\bigcup_{i \in Q_{\mathcal{B}}} \hat{\mathcal{P}}_i \cup \hat{\mathcal{H}}_i\}$. Interestingly, due to the multi-positive nature of XMC, sampled hard-negatives for $\mathbf{x}_i$ might turn out to be an unsampled positive label for $\mathbf{x}_j$. More formally, $\exists \; j \in Q_{\mathcal{B}} \; s.t. \; \{\hat{\mathcal{H}}_i \cap \mathcal{P}_j \neq \phi, \; \hat{\mathcal{H}}_i \cap \mathcal{P}_j^{\mathcal{B}} = \phi\}$. This requires altering the definition of $\mathcal{P}_i^{\mathcal{B}}$ to accommodate these *extra* positives (represented by the dark green square in Figure 2) as $\mathcal{P}_i^{\mathcal{B}} = \{\hat{\mathcal{P}}_i \cup \{\bigcup_{j \in \{Q_{\mathcal{B}} - \{i\}\}} \{\hat{\mathcal{P}}_j \cup \hat{\mathcal{H}}_j\} \cap \mathcal{P}_i\}\}$. This effect is also quantified in Figure 2b. Query clustering for batching and dynamic ANNS hard-negative mining strategies complement each other, since the presence of similar queries leads to a higher overlap in their positives *and hard negatives*, enabling us to scale the effective size of the label pool. Further, to provide $\Phi_{\mathfrak{D}}$ and $\Phi_{\mathfrak{C}}$ with progressively harder negatives, the ANNS is refreshed every $\tau$ epochs and to uniformly sample hard negatives, we keep $|\mathcal{H}| = \eta \times \tau$.

---

**Algorithm 1** Training step in UNIDEC

---

**Input:** instance $\mathbf{x}$, label features $\mathbf{z}$, positive labels $\mathcal{P}$, encoder $\Phi$, classifier lookup-table $\Psi$, non-linear transformations $g_1(\cdot)$ and $g_2(\cdot)$ $\Phi_{\mathfrak{D}}(\cdot)$, $\Phi_{\mathfrak{C}}(\cdot) := \mathfrak{N}(g_1(\Phi(\cdot)))$, $g_2(\Phi(\cdot))$ **for** $e$ in $1..e$ **do**

 **if** $e \% \tau$ is 0 **then**

  $\mathfrak{B} \leftarrow \text{CLUSTER}(\Phi_{\mathfrak{D}}(\mathbf{x}_i)|_{i=0}^{N})$   $\mathcal{H} \leftarrow \text{topk}(\text{ANNS}(\Phi_{\mathfrak{D}}(\mathbf{x}_i)|_{i=0}^{N}, \Phi_{\mathfrak{D}}(\mathbf{z}_l)|_{l=0}^{L}))$

 **for** $Q_{\mathcal{B}}$ *in* $\mathfrak{B}$ **do**

  **for** $i$ *in* $Q_{\mathcal{B}}$ **do**

   $\hat{\mathcal{P}}_i \leftarrow \text{sample}(\mathcal{P}_i, \beta)$

   $\hat{\mathcal{H}}_i \leftarrow \text{sample}(\mathcal{H}_i - \mathcal{P}_i, \eta)$

  $L_{\mathcal{B}} \leftarrow \{\bigcup_{i \in Q_{\mathcal{B}}} \hat{\mathcal{P}}_i \cup \hat{\mathcal{H}}_i\}$

  $\mathcal{P}^{\mathcal{B}} \leftarrow \{\{\mathcal{P}_i \cap L_{\mathcal{B}}\}|_{i \in Q_{\mathcal{B}}}\}$

  $\mathcal{P}^{L} \leftarrow \{\{i \, | \, i \in Q_{\mathcal{B}}, \, \mathcal{P}_{i,l} = 1\}|_{l \in L_{\mathcal{B}}}\}$

  $\mathcal{L}_{\mathfrak{D},q2l} \leftarrow \mathcal{L}_{PSL}(\Phi_{\mathfrak{D}}(\mathbf{x}_i), \Phi_{\mathfrak{D}}(\mathbf{z}_l) \mid \mathcal{B}, \mathcal{P}^{\mathcal{B}})$

  $\mathcal{L}_{\mathfrak{D},l2q} \leftarrow \mathcal{L}_{PSL}(\Phi_{\mathfrak{D}}(\mathbf{z}_l), \Phi_{\mathfrak{D}}(\mathbf{x}_i) \mid \mathcal{B}, \mathcal{P}^{L})$

  $\mathcal{L}_{\mathfrak{D}} \leftarrow \lambda_{\mathfrak{D}} \cdot \mathcal{L}_{\mathfrak{D},q2l} + (1 - \lambda_{\mathfrak{D}}) \cdot \mathcal{L}_{\mathfrak{D},l2q}$

  $\mathcal{L}_{\mathfrak{C},q2l} \leftarrow \mathcal{L}_{PSL}(\Phi_{\mathfrak{C}}(\mathbf{x}_i), \Psi(l) \mid \mathcal{B}, \mathcal{P}^{\mathcal{B}})$

  $\mathcal{L}_{\mathfrak{C},l2q} \leftarrow \mathcal{L}_{PSL}(\Psi(l), \Phi_{\mathfrak{C}}(\mathbf{x}_i) \mid \mathcal{B}, \mathcal{P}^{L})$

  $\mathcal{L}_{\mathfrak{C}} \leftarrow \lambda_{\mathfrak{C}} \cdot \mathcal{L}_{\mathfrak{C},q2l} + (1 - \lambda_{\mathfrak{C}}) \cdot \mathcal{L}_{\mathfrak{C},l2q}$

  $\mathcal{L} \leftarrow \lambda \cdot \mathcal{L}_{\mathfrak{D}} + (1 - \lambda) \cdot \mathcal{L}_{\mathfrak{C}}$

  adjust $\Phi$, $g_1(\cdot)$, $g_2(\cdot)$ and $\Psi$ to reduce loss $\mathcal{L}$.

---

Note that $L_{\mathfrak{D},q2l}$ denotes the multi-class loss between $\mathbf{x}_i$ and $\mathbf{z}_l \; \forall \; l \in L_{\mathcal{B}}$. As the data points and labels in XMC tasks belong to the same vocabulary universe (such as product recommendation), we find it beneficial to optimize $L_{\mathfrak{D},l2q}$ alongside $L_{\mathfrak{D},q2l}$, making $\mathcal{L}_{\mathfrak{D}}$ a symmetric loss. Since [29], a plethora of works have leveraged symmetric optimizations in the vision-language retrieval pre-training domain. For XMC, the interchangability of $Q_{\mathcal{B}}$ and $L_{\mathcal{B}}$ in the symmetric objective can be viewed equivalent to (i) feeding more data relations in a batch, and (ii) bridging missing relations in the dataset [20]. Further, we formulate XMC as a symmetric problem from $L_{\mathcal{B}}$ to $Q_{\mathcal{B}}$, thus calculating the multi-class loss between

$\mathbf{z}_l$ and $\mathbf{x}_i \; \forall \; i \in Q_{\mathcal{B}}$ given by:

$$\mathcal{L}_{\mathfrak{D},l2q} = \mathcal{L}_{\text{PSL}}(\Phi_{\mathfrak{D}}(z_l), \Phi_{\mathfrak{D}}(\mathbf{x}) \mid \mathcal{B}, \mathcal{P}^{L})$$

Note that, $\mathcal{P}^{L} = \{i \mid i \in Q_{\mathcal{B}}, \; l \in L_{\mathcal{B}}, \; \mathcal{P}_{i,l} = 1\}$. The total DE contrastive loss can thus be written as (note, for simplicity we use $\lambda_{\mathfrak{D}} = 0.5$ for all datasets, which works well in practice):

$$\mathcal{L}_{\mathfrak{D}} = \lambda_{\mathfrak{D}} \cdot \mathcal{L}_{\mathfrak{D},q2l} + (1 - \lambda_{\mathfrak{D}}) \cdot \mathcal{L}_{\mathfrak{D},l2q}$$

## 3.3 Unified Classifier Training with Pick-some-Labels

XMC classifiers are trained on a shortlist consisting of *all* positive and $O(Log(L))$ hard negative labels [7]. As the reader can observe from Figure 1 and Algorithm 1, the document and label embedding computation and batch pool is shared between $\Phi_{\mathfrak{D}}$ and $\Phi_{\mathfrak{C}}$. We simply unify the classifier training with that of DE by leveraging the same *PSL* reduction used for contrastive learning, with only minor changes: $\Phi_{\mathfrak{C}}(\mathbf{x}_i)|_{i \in Q_{\mathcal{B}}}$ replaces $\Phi_{\mathfrak{D}}(\mathbf{x}_i)|_{i \in Q_{\mathcal{B}}}$ and, the label embeddings $\Phi_{\mathfrak{D}}(\mathbf{z}_l)|_{l \in L_{\mathcal{B}}}$ are replaced by $\Psi_l|_{l \in L_{\mathcal{B}}}$. Formally, the multi-class *PSL* loss for classifier $L_{\mathfrak{C},q2l}$ can be defined as:

$$\mathcal{L}_{\mathfrak{C},q2l} = \mathcal{L}_{\text{PSL}}(\Phi_{\mathfrak{C}}(\mathbf{x}), \Psi_l \mid \mathcal{B}, \mathcal{P}^{\mathcal{B}})$$

Similar to DE training, we find it beneficial to employ a symmetric loss for classifier training, defined (with $\lambda_{\mathfrak{C}} = 0.5$) as:

$$\mathcal{L}_{\mathfrak{C}} = \lambda_{\mathfrak{C}} \cdot \mathcal{L}_{\mathfrak{C},q2l} + (1 - \lambda_{\mathfrak{C}}) \cdot \mathcal{L}_{\mathfrak{C},l2q}$$

Finally, we combine the two losses and train together in an end-to-end fashion, thereby achieving Unification of DE and classifier training for XMC.

$$\mathcal{L} = \lambda \mathcal{L}_{\mathfrak{D}} + (1 - \lambda) \mathcal{L}_{\mathfrak{C}}$$

## 3.4 Inference

For ANNS inference, the label graph can either be created over the encoded label embeddings $\{\Phi_{\mathfrak{D}}(z_l)\}_{l=1}^{L}$ or the label classifier embeddings $\{\mathfrak{N}(\Psi(l))\}_{l=1}^{L}$, which are queried by $\{\Phi_{\mathfrak{D}}(\mathbf{x}_i)\}_{i=1}^{N}$ or $\{\mathfrak{N}(\Phi_{\mathfrak{C}}(\mathbf{x}_i))\}_{i=1}^{N}$ respectively. Even though we train the classifiers over an un-normalized embedding space, we find it empirically beneficial to perform ANNS search over the unit normalized embedding space [10, 21]. Interestingly, the concatenation of these two embeddings leads to a much more efficient retrieval. More specifically, we create the ANNS retrieval graph over the concatenated label representation $\{\Phi_{\mathfrak{D}}(z_l) \oplus \mathfrak{N}(\Psi(l))\}_{l=0}^{L}$, which is queried by the concatenated document representations $\{\Phi_{\mathfrak{D}}(\mathbf{x}_i) \oplus \mathfrak{N}(\Phi_{\mathfrak{C}}(\mathbf{x}_i))\}_{i=0}^{N}$. Intuitively, this is a straight-forward way to ensemble the similarity scores from both the embedding spaces.

## 4 Experiments

**Datasets:** We benchmark our experiments on 6 standard datasets, comprising of both long-text inputs (LF-Amazon-131K, LF-WikiSeeAlso-320K) and short-text inputs (LF-AmazonTitles-131K, LF-AmazonTitles-1.3M, LF-WikiTitles-500K, LF-WikiSeeAlsoTitles-320K). We also evaluate baselines on a proprietary *Query2Bid* dataset, comprising of 450M labels, which is orders of magnitude larger than any public dataset. Details of these datasets can be found at [3] and in Appendix B.

***Baselines & Evaluation Metrics:*** We compare against two classes of Baselines namely, (i) DE Approaches ($\Phi$) consisting of only an encoder [6, 11, 16, 36] and, (ii) Classifier Based Approaches ($\Psi$) which use linear classifiers, with or without the encoder [6, 13]. A more comprehensive comparison with baselines has been provided in Appendix C. We use popular metrics such as Precision@K and Propensity-scored Precision@K (K $\in \{1, 3, 5\}$), defined in [3].

***Implementation Details and Ablation Study.*** We initialize both DePSL, our purely dual encoder method and UniDEC with a pre-trained 6L-Distilbert and train the $\Phi$, $g(\cdot)$ and $\Psi$ with a learning rate of $1e-4$, $2e-4$ and $1e-3$ respectively using cosine annealing with warm-up as the scheduler, hard-negative shortlist refreshed every $\tau = 5$ epochs. We make an effort to minimize the role of hyperparameters by keeping them almost same across all datasets.

## 4.1 Evaluation on XMC Datasets

In these settings, we evaluate DePSL and UniDEC against both DE and XMC baselines. UniDEC differs from these baselines in the following ways, (i) on training objective, UniDEC uses the proposed *PSL* relaxation of PAL for both DE and CLF training, instead of POL reduction used by existing methods like Ngame and Dexml, (ii) UniDEC does away with the need of modular training by unifying DE and CLF, (iii) finally, UniDEC framework adds explicitly mined hard negatives to the negative mining-aware batches which helps increase P@K metrics (see Table 5).

***UniDEC/DePSL vs Ngame($\Phi$):*** Table 1 depicts that UniDEC ($\Phi \oplus \Psi$) consistently outperforms Ngame($\Psi$) (it's direct comparison baseline), where we see gains of $2-8\%$ in P@K and upto 10% on PSP@K. DePSL, on the other hand, outperforms Ngame on P@k with improvements ranging from $2-9\%$. For PSP@k, DePSL ($\Phi$) always outperforms Ngame ($\Phi$) on long-text datasets, while the results are mixed on short-text datasets.

***DePSL vs DPR/ANCE:.*** Empirical performance of DPR demonstrates the limits of a DE model trained with InfoNCE loss and random in-batch negatives (popular in DR methods). Evidently, Ance improves over DPR in the P@K metrics, which can be observed as the impact of explicitly mining hard-negative labels per instance instead of solely relying on the random in-batch negatives. Even though, these approaches use 12L-Bert-base instead of 6L-Distilbert common in XMC methods, Ance only shows marginal gains over Ngame on both datasets. Our proposed DE method, DePSL, despite using half the # Layers and half the search embedding dimension, is able to surpass these DR approaches by $15-20\%$ for P@K metrics over LF-AmazonTitles-1.3M dataset.

***Search Dimensionality.*** As mentioned before, DePSL outperforms Ngame on P@K metrics across benchmarks. Notably, DePSL does so by projecting (using $g_1(\cdot)$) and training the encoder embeddings in a low-dimension space of $d = 384$. Similarly, for UniDEC, inference is carried out by concatenating $\mathfrak{R}(\Phi_{\mathfrak{C}})$ and $\Phi_{\mathfrak{D}}$ embeddings. Here, both $g_1(\cdot)$ and $g_2(\cdot)$ consist of linear layers projecting $\Phi(\cdot)$ into a low-dimensional space of $d = 256$ or $d = 384$. On the other hand, all aforementioned baselines use a higher dimension of 768 for both DE and CLF evaluations. For the proprietary Query2Bid-450M dataset, we use final dimension of 64 for all the methods necessitated by constraints of online serving.

***Applicability to Real-World Data:*** Finally, we also demonstrate the applicability of DePSL to real-world sponsored search dataset, Query2Bid-450M in Appendix A, where it is observed to be $\sim 1.5\%$ better in P@K than leading DR & XMC methods. Additionally, DePSL was deployed on a live search engine where A/B tests indicated that it improved popular metrics such as IY, CY, CTR, QC over an ensemble of DR and XMC techniques by 0.87%, 0.66%, 0.21% and 1.11% respectively.

## 4.2 Efficiency Comparison with Spectrum of XMC methods

In this section, we provide a comprehensive comparison (refer Table 2) of our proposed DePSL and UniDEC with two extreme ends of XMC spectrum (refer Figure 1): (i) Renee, which is initialized with pre-trained Ngame encoder, trains OvA classifiers with the BCE loss and, (ii) DEXML which achieves SOTA performance by training a DE using their proposed loss function *decoupled softmax*. Note that, these approaches do not pose a fair comparison with our proposed approaches as both Renee and Dexml do not use a label shortlist and backpropagate over the entire label space, requiring an order of magnitude higher GPU VRAM to run an iteration on LF-AmazonTitles-1.3M. Therefore, for the same encoder, they can be considered as the upper bound of empirical performance of CLF (OvA) and DE methods respectively. Table 2 shows that similar, and perhaps better, performance is possible by using our proposed UniDEC and leveraging the proposed *PSL* reduction of multi-class losses over a label shortlist.

***Comparison with Renee :*** We observe that UniDEC delivers matching performance over P@K and PSP@K metrics on long-text datasets and significantly outperforms Renee on LF-AmazonTitles-1.3M. In fact, our proposed DE method outperforms Renee on LF-Wikipedia-500K without even employing classifiers. We posit that UniDEC is therefore more effective for skewed datasets, with higher avg. points per label and more tail labels. Furthermore, these observations imply while Renee helps BCE loss reach it's empirical limits by scaling over the entire label space, with the UniDEC framework, we can match this limit with a shortlist that is $86-212\times$ smaller than the label space, thereby consuming significantly lower compute ($1 \times$ A6000 vs $8 \times$ V100).

***DePSL vs Dexml (with shortlist):*** While DePSL leverages the proposed *PSL* reduction in the UniDEC framework, the latter uses the POL reduction with the same loss function. As evident in the LF-AmazonTitles-1.3M, Table 2, (i) For a comparable label pool size (4000 vs 8192), DePSL significantly outperforms DEXML by $\sim 20\%$ in P@K metrics. (ii) To achieve similar performance as DePSL, DEXML need to use an effective label pool size of 90K. However in the same setting, DePSL needs only $1/4^{th}$ batch size and $1/22^{th}$ label pool size. A similar trend is seen in LF-Wikipedia-500K. These observations empirically demonstrate the informativeness of the batches in UniDEC - the same information can be captured by it with significantly smaller batch sizes.

| Method | $d$ | P@1 | P@3 | P@5 | PSP@1 | PSP@3 | PSP@5 | $d$ | P@1 | P@3 | P@5 | PSP@1 | PSP@3 | PSP@5 |
|---|---|---|---|---|---|---|---|---|---|---|---|---|---|---|
| *Long-text →* | | | | | LF-Amazon-131K | | | | | | | LF-WikiSeeAlso-320K | | |
| Ngame (Φ) | 768 | 42.61 | 28.86 | 20.69 | **38.27** | 43.75 | 48.71 | 768 | 43.58 | 28.01 | 20.86 | 30.59 | 33.29 | 36.03 |
| DePSL (Φ) | 512 | **45.86** | **30.52** | **21.89** | 38.19 | **44.07** | **49.56** | 512 | **44.83** | 29.07 | **21.66** | **30.67** | 33.56 | 36.41 |
| SiameseXML (Ψ) | 300 | 44.81 | - | 21.94 | 37.56 | 43.69 | 49.75 | 300 | 42.16 | - | 21.35 | 29.01 | 32.68 | 36.03 |
| Ngame (Ψ) | 768 | 46.95 | 30.95 | 22.03 | 38.67 | 44.85 | 50.12 | 768 | 45.74 | 29.61 | 22.07 | 30.38 | 33.89 | 36.95 |
| UniDEC (Φ ⊕ Ψ) | 768 | **47.80** | **32.29** | **23.35** | **40.28** | **47.03** | **53.24** | 768 | **47.69** | **30.74** | **22.81** | **35.45** | **38.02** | **40.71** |
| *Short-text →* | | | | | LF-WikiTitles-500K | | | | | | | LF-WikiSeeAlsoTitles-320K | | |
| GraphSage (Φ) | 768 | 27.30 | 17.17 | 12.96 | 21.56 | 21.84 | 23.50 | 768 | 27.19 | 15.66 | 11.30 | 22.35 | 19.31 | 19.15 |
| Ngame (Φ) | 768 | 29.68 | 18.06 | 12.51 | 23.18 | 22.08 | 21.18 | 768 | 30.79 | 20.34 | 15.36 | **25.14** | **26.77** | **28.73** |
| DePSL (Φ) | 512 | **49.66** | **27.93** | **19.62** | **27.44** | **25.64** | **24.94** | 512 | **33.91** | **21.92** | **16.48** | 24.22 | 25.80 | 27.99 |
| Ngame (Ψ) | 768 | 39.04 | 23.10 | 16.08 | 23.12 | 23.31 | 23.03 | 768 | 32.64 | 22.00 | 16.60 | 24.41 | 27.37 | 29.87 |
| CascadeXML (Ψ) | 768 | 47.29 | 26.77 | 19.00 | 19.19 | 19.47 | 19.75 | 768 | 23.39 | 15.71 | 12.06 | 12.68 | 15.37 | 17.63 |
| UniDEC (Φ ⊕ Ψ) | 768 | **50.22** | **28.76** | **20.32** | **25.90** | **25.20** | 24.85 | 768 | **36.28** | **23.23** | **17.31** | **26.31** | **27.81** | **29.90** |
| *Short-text →* | | | | | LF-AmazonTitles-131K | | | | | | | LF-AmazonTitles-1.3M | | |
| DPR(Φ) | 768 | 41.85 | 28.71 | 20.88 | 38.17 | 43.93 | 49.45 | 768 | 44.64 | 39.05 | 34.83 | 32.62 | 35.37 | 36.72 |
| ANCE (Φ) | 768 | 42.67 | 29.05 | 20.98 | 38.16 | 43.78 | 49.03 | 768 | 46.44 | 41.48 | 37.59 | 31.91 | 35.31 | **37.25** |
| Ngame (Φ) | 768 | **42.61** | 28.86 | 20.69 | **38.27** | **43.75** | **48.71** | 768 | 45.82 | 39.94 | 35.48 | **33.03** | **35.63** | 36.80 |
| DePSL (Φ) | 512 | 42.34 | **28.98** | **20.87** | 37.61 | 43.01 | 47.93 | 384 | **54.20** | **48.20** | **43.38** | 30.17 | 34.11 | 36.25 |
| SiameseXML (Ψ) | 300 | 41.42 | 30.19 | 21.21 | 35.80 | 40.96 | 46.19 | 300 | 49.02 | 42.72 | 38.52 | 27.12 | 30.43 | 32.52 |
| Ngame (Ψ) | 768 | **44.95** | 29.87 | **21.20** | 38.25 | 43.75 | 48.42 | 768 | 54.69 | 47.76 | 42.80 | 28.23 | 32.26 | 34.48 |
| UniDEC (Φ ⊕ Ψ) | 768 | 44.35 | 29.49 | 21.03 | **39.23** | **44.13** | **48.90** | 512 | **57.41** | **50.75** | **45.89** | **30.10** | **34.32** | **36.78** |

Table 1: Experimental results showing the effectiveness of DePSL and UniDEC against both state-of-the-art dual encoder approaches and extreme classifiers. The best-performing results are put in bold. DE and classifier results are compared separately.

**UniDEC vs DEXML-Full:** UniDEC, scales to LF-AmazonTitles-1.3M on a single A6000 GPU using a label shortlist of only 3000 labels, as opposed to DEXML-Full which requires 16 A100s and uses the entire label space of 1.3M. Despite this, Table 2 indicates that UniDEC matches DEXML-Full on P@5 and PSP@5 metrics.

## 4.3 Ablation Study

**Evaluation with Multiple Loss Functions :** As mentioned previously, any loss function can be chosen in the UniDEC framework, however, we experiment with two multi-class losses in particular, namely *SupCon* loss (SC) [21] and *Decoupled Softmax* (DS) [11]. Replacing $\ell_{MC}$ with these gives

$$\mathcal{L}_{SC} = \sum_{i \in Q_{\mathcal{B}}} \frac{-1}{|\mathcal{P}_i^{\mathcal{B}}|} \sum_{p \in \mathcal{P}_i^{\mathcal{B}}} \log \frac{\exp(\langle \Phi_{\mathfrak{D}}(\mathbf{x}_i), \Phi_{\mathfrak{D}}(\mathbf{z}_p) \rangle / \tau)}{\sum\limits_{l \in L_{\mathcal{B}}} \exp(\langle \Phi_{\mathfrak{D}}(\mathbf{x}_i), \Phi_{\mathfrak{D}}(\mathbf{z}_l) \rangle / \tau)}$$

$$\mathcal{L}_{DS} = \sum_{i \in Q_{\mathcal{B}}} \frac{-1}{|\mathcal{P}_i^{\mathcal{B}}|} \sum_{p \in \mathcal{P}_i^{\mathcal{B}}} \log \frac{\exp(\langle \Phi_{\mathfrak{D}}(\mathbf{x}_i), \Phi_{\mathfrak{D}}(\mathbf{z}_p) \rangle / \tau)}{\sum\limits_{l \in L_{\mathcal{B}} / \{\mathcal{P}_i^{\mathcal{B}} - p\}} \exp(\langle \Phi_{\mathfrak{D}}(\mathbf{x}_i), \Phi_{\mathfrak{D}}(\mathbf{z}_l) \rangle / \tau)}$$

Notably, from Table 3 and Table 4, we observe that *Decoupled Softmax* turns out to be a better loss for XMC tasks as it helps the logits scale better [11] as compared to *SupCon* which caps the gradient due to a hard requirement of producing a probability distribution. We further observe that classifier performance can further improve by adding BCE loss as an auxiliary OvA loss to the classifier loss. While this helps enhance P@K metrics, the PSP@K metrics take a significant dip on the inclusion of auxiliary BCE loss.

These observations are in line with the performance of *Renee* which leverages BCE loss and suffers on PSP@K metrics. Simply using BCE loss for classifier works in our pipeline, however, ends up performing worse than using multi-class loss to train the classifiers.

**UniDEC Framework :** We show the effect of the two individual components $\Phi_{\mathfrak{D}}$ and $\Phi_{\mathfrak{C}}$ of UniDEC in Table 5. The scores are representative of the evaluation of the respective component of the UniDEC framework, (i) UniDEC-DE ($\Phi_{\mathfrak{D}}$) performs inference with an ANNS built over $\Phi_{\mathfrak{D}}(\mathbf{z}_l)|_{l=0}^L$, (ii) UniDEC-CLF ($\Phi_{\mathfrak{C}}$) performs inference with an ANNS built over $\mathfrak{R}(\Psi(l))|_{l=0}^L$ and (iii) UniDEC uses the ANNS built over the concatenation of both $\{\Phi_{\mathfrak{D}}(\mathbf{z}_l) + \mathfrak{R}(\Psi(l))\}|_{l=0}^L$. Notably, concatenation of embeddings leads to a more effective retrieval. We attribute its performance to two aspects, (i) as seen in previous XMC models, independent classifier weights significantly improve the discriminative capabilities of these models and (ii) we hypothesise that normalized and unnormalized spaces learn complementary information which leads to enhanced performance when an ANNS is created on their aggregation. Note that, the individual search dimensions of UniDEC-DE : and UniDEC-CLF are $d/2$ and searching with a concatenated embedding leads to a fair comparison with other baselines which use a dimensionality of $d$. Note that for all experiments in the paper, $g_1(\cdot), g_2(\cdot)$ is defined

| Method | P@1 | P@5 | PSP@1 | PSP@5 | $|Q_\mathcal{B}|$ | $|L_\mathcal{B}|$ | VRAM | TT |
|---|---|---|---|---|---|---|---|---|
| *w Classifiers* | | | LF-Amazon-131K | | | | | |
| Renee | **48.05** | 23.26 | 39.32 | **53.51** | 512 | 131K | 128 | 58 |
| UniDEC | 47.80 | **23.35** | **40.28** | 53.24 | 576 | 3000 | **48** | 24 |
| *w Classifiers* | | | LF-WikiSeeAlso-320K | | | | | |
| Renee | **47.70** | **23.82** | 31.13 | 40.37 | 2048 | 320K | 128 | 81 |
| UniDEC | 47.69 | 22.81 | **35.45** | **40.71** | 677 | 3500 | **48** | 39 |
| | | | LF-Wikipedia-500K | | | | | |
| DEXML | 77.71 | 43.32 | - | - | 2048 | 2048 | 80 | - |
| DEXML | 84.77 | **50.31** | - | - | 2048 | 22528 | 160 | - |
| DePSL | **85.20** | 49.88 | 45.96 | 59.31 | 221 | 3000 | **48** | 55 |
| Renee | 84.95 | 51.68 | 39.89 | 56.70 | 2048 | 500K | 320 | 39 |
| DEXML-Full | 85.78 | 50.53 | 46.27 | 58.97 | 2048 | 500K | 320 | 39 |
| *Dual Encoder* | | | LF-AmazonTitles-1.3M | | | | | |
| DEXML | 42.15 | 32.97 | - | - | 8192 | 8192 | 160 | - |
| DEXML | 54.01 | 42.08 | 28.64 | 33.58 | 8192 | 90112 | 320 | - |
| DePSL | **54.20** | **43.38** | 30.17 | 36.25 | 2200 | 4000 | **48** | 53 |
| Renee | 56.04 | 45.32 | 28.56 | 36.14 | 1024 | 1.3M | 256 | 105 |
| UniDEC | **57.41** | **45.89** | 30.10 | 36.78 | 1098 | 3000 | **48** | 78 |
| DEXML-Full | 58.40 | 45.46 | 31.36 | 36.58 | 8192 | 1.3M | 640 | 66 |

**Table 2: Experimental results showing the effectiveness of DePSL and UniDEC against the two ends of XMC spectrum. $|Q_\mathcal{B}|$ denotes batch size, $|L_\mathcal{B}|$ denotes label pool size and TT denotes Training Time(in hrs). Note, these comparisons are not fair owing to the significant gap in used resources.**

| | LF-AmazonTitles-1.3M | | | | LF-WikiTitles-500K | | | |
|---|---|---|---|---|---|---|---|---|
| Method | P@1 | P@5 | PSP@1 | PSP@5 | P@1 | P@5 | PSP@1 | PSP@5 |
| DE loss - SupCon; CLF loss - SupCon | | | | | | | | |
| UniDEC | 53.41 | 43.57 | 32.54 | 38.20 | 48.38 | 19.89 | 26.26 | 24.78 |
| UniDEC-de | 49.35 | 39.23 | 27.78 | 32.86 | 48.63 | 19.30 | 27.21 | 24.52 |
| UniDEC-clf | 46.90 | 38.62 | 31.23 | 35.28 | 29.48 | 14.21 | 18.97 | 18.82 |
| DE loss - SupCon; CLF loss - SupCon + BCE | | | | | | | | |
| UniDEC | 54.86 | 44.61 | 28.05 | 35.05 | 49.68 | 20.15 | 25.29 | 24.67 |
| UniDEC-de | 51.08 | 40.78 | 28.64 | 34.00 | 47.07 | 18.88 | 27.47 | 24.53 |
| UniDEC-clf | 53.48 | 42.76 | 26.24 | 32.81 | 45.07 | 17.96 | 19.01 | 19.68 |
| DE loss - Decoupled Softmax; CLF loss - BCE | | | | | | | | |
| UniDEC | 55.12 | 44.80 | 31.72 | 37.28 | 48.65 | 19.58 | 26.15 | 24.37 |
| UniDEC-de | 50.83 | 40.61 | 27.14 | 33.66 | 47.70 | 18.59 | 26.84 | 24.19 |
| UniDEC-clf | 54.69 | 42.81 | 30.74 | 36.48 | 43.27 | 16.55 | 19.41 | 18.29 |
| DE loss - Decoupled Softmax; CLF loss - Decoupled Softmax | | | | | | | | |
| UniDEC | 56.73 | 45.19 | **34.03** | 39.54 | 48.97 | 19.82 | **27.08** | 24.89 |
| UniDEC-de | 52.52 | 42.02 | 29.78 | 35.06 | 49.20 | 19.30 | 27.36 | 24.49 |
| UniDEC-clf | 43.67 | 36.85 | 32.68 | 37.07 | 30.27 | 13.74 | 18.95 | 17.75 |
| DE loss - Decoupled Softmax; CLF loss - Decoupled Softmax + BCE | | | | | | | | |
| UniDEC | **57.41** | **45.89** | 30.10 | 36.78 | **50.22** | **20.32** | 25.90 | 24.85 |
| UniDEC-de | 52.51 | 42.00 | 29.82 | 35.08 | 49.16 | 19.33 | 27.35 | 24.54 |
| UniDEC-clf | 55.56 | 44.10 | 29.15 | 35.49 | 44.66 | 17.38 | 20.56 | 19.62 |

**Table 3: Experimental results showing the effect of different loss functions while training UniDEC. Further, the table also shows the scores of inference done using only the DE head $\Phi_\mathfrak{D}(x)$ or the normalized CLF head $\mathfrak{R}(\Phi_\mathfrak{C}(x))$, instead of the concatenated vector.**

| Loss | Dual | P@1 | P@3 | P@5 | PSP@1 | PSP@3 | PSP@5 | P@1 | P@3 | P@5 | PSP@1 | PSP@3 | PSP@5 |
|---|---|---|---|---|---|---|---|---|---|---|---|---|---|
| | | | | | LF-Amazon-131K | | | | | | LF-WikiSeeAlso-320K | | |
| SC | | 44.41 | 29.84 | 21.63 | 36.89 | 43.00 | 48.90 | 43.79 | 28.31 | 21.08 | 29.47 | 32.15 | 34.89 |
| SC | ✓ | 45.03 | 30.24 | 21.93 | 37.66 | 43.81 | 49.78 | 44.32 | 28.84 | 21.57 | 30.16 | 33.14 | 36.11 |
| DS | | 45.86 | 30.52 | 21.89 | 38.19 | 44.07 | 49.56 | 44.73 | 28.78 | 21.43 | 30.18 | 32.79 | 35.58 |
| DS | ✓ | **45.79** | **30.60** | **21.95** | **38.43** | **44.42** | **49.92** | **44.83** | **29.07** | **21.66** | **30.67** | **33.56** | **36.41** |
| | | | | | LF-AmazonTitles-1.3M | | | | | | LF-WikiTitles-500K | | |
| SC | | 52.22 | 46.45 | 41.80 | 29.15 | 32.90 | 34.91 | 48.30 | 27.33 | 19.26 | 27.00 | 25.12 | 24.51 |
| SC | ✓ | 50.62 | 45.09 | 40.64 | 30.51 | 34.30 | 36.33 | 47.33 | 26.79 | 18.94 | 27.41 | 25.21 | 24.63 |
| DS | | **54.20** | **48.20** | **43.38** | 30.17 | 34.11 | 36.25 | **49.66** | **27.93** | **19.62** | 27.44 | 25.64 | 24.94 |
| DS | ✓ | 53.26 | 47.55 | 42.86 | **31.90** | **35.80** | **37.88** | 48.87 | 27.47 | 19.35 | **28.09** | **25.77** | **25.08** |

**Table 4: Experimental results showing the effect of adding dual loss while training our DePSL.**

as follows,

$$\Phi_\mathfrak{D}(\cdot),\ \Phi_\mathfrak{C}(\cdot) \coloneqq \mathfrak{R}(g_1(\Phi(\cdot))),\ g_2(\Phi(\cdot))$$

$$g_1(\cdot) \coloneqq \mathrm{nn.\,Sequential(nn.\,Linear}(d_\Phi, d),\, \mathrm{nn.\,Tanh()},\, \mathrm{nn.\,Dropout}(0.1))$$

$$g_2(\cdot) \coloneqq \mathrm{nn.\,Sequential(nn.\,Linear}(d_\Phi, d),\, \mathrm{nn.\,Dropout}(0.1))$$

**Effect of ANNS-mind Hard Negatives :** The effect of explicitly adding ANNS-mined hard negatives is shown via a vis-a-vis comparison with UniDEC **(w/o Hard Negatives)** in Table 5. Here, when we do not add hard negatives, we compensate by adding other positives of the batched queries. More broadly, we observe a P vs PSP trade-off in this ablation. We find that not including hard negatives in the shortlist performs better on PSP@K metrics, due to inclusion of more positive labels. Consequently, adding (typically $\eta = 6$) hard negatives generally increases performance on P@K metrics, while compromising on PSP@K metrics. While the smaller datasets show only marginal improvements with added hard negatives, these effects are more pronounced in the larger datasets, proving its necessity in the pipeline.

**Effect of Symmetric Loss.** As shown in Table 4, making the loss function symmetric has a favorable effect on all metrics for *long-text* datasets. However, this make the *short-text* datasets favor PSP@K metrics more, at an expense of P@K metrics. We believe this happens because of mixing data distributions. While adding a

| Method | P@1 | P@3 | P@5 | PSP@1 | PSP@3 | PSP@5 | P@1 | P@3 | P@5 | PSP@1 | PSP@3 | PSP@5 |
|---|---|---|---|---|---|---|---|---|---|---|---|---|
| | | | LF-Amazon-131K | | | | | | | w/o Hard Negatives | | |
| UniDEC | 47.80 | 32.29 | 23.35 | 40.28 | 47.03 | 53.24 | 47.45 | 31.49 | 22.53 | 41.28 | 46.93 | 52.40 |
| UniDEC-de | 45.24 | 30.32 | 21.97 | 37.85 | 43.92 | 49.90 | 45.45 | 30.29 | 21.79 | 38.15 | 43.97 | 49.53 |
| UniDEC-clf | 47.83 | 32.31 | 23.32 | 40.56 | 47.17 | 53.21 | 43.77 | 28.30 | 19.90 | 39.80 | 43.64 | 47.78 |
| | | | LF-WikiSeeAlso-320K | | | | | | | w/o Hard Negatives | | |
| UniDEC | 47.69 | 30.74 | 22.81 | 35.45 | 38.02 | 40.71 | 46.74 | 30.04 | 22.27 | 35.85 | 38.10 | 40.97 |
| UniDEC-de | 44.65 | 28.84 | 21.53 | 30.54 | 33.30 | 36.20 | 43.07 | 27.67 | 20.69 | 30.52 | 32.81 | 35.55 |
| UniDEC-clf | 42.04 | 25.89 | 18.89 | 33.63 | 34.07 | 35.60 | 41.05 | 25.47 | 18.80 | 33.52 | 34.01 | 35.82 |
| | | | LF-WikiTitles-500K | | | | | | | w/o Hard Negatives | | |
| UniDEC | 50.22 | 28.76 | 20.32 | 25.90 | 25.20 | 24.85 | 49.84 | 28.31 | 19.95 | 26.41 | 25.44 | 24.99 |
| UniDEC-de | 49.16 | 27.51 | 19.33 | 27.35 | 25.24 | 24.54 | 46.87 | 25.60 | 17.83 | 27.38 | 24.64 | 23.81 |
| UniDEC-clf | 44.66 | 24.81 | 17.38 | 20.56 | 19.86 | 19.62 | 44.20 | 24.83 | 17.48 | 21.33 | 20.60 | 20.42 |
| | | | LF-AmazonTitles-1.3M | | | | | | | w/o Hard Negatives | | |
| UniDEC | 57.41 | 50.75 | 45.89 | 30.10 | 34.32 | 36.78 | 56.51 | 49.77 | 44.94 | 31.90 | 35.89 | 38.21 |
| UniDEC-de | 52.51 | 46.66 | 42.00 | 29.82 | 33.21 | 35.08 | 49.71 | 43.87 | 39.37 | 30.40 | 33.49 | 35.20 |
| UniDEC-clf | 55.56 | 48.77 | 44.10 | 29.15 | 33.15 | 35.49 | 53.31 | 47.21 | 42.90 | 30.41 | 34.19 | 36.47 |

Table 5: Experimental results showing the effect of adding ANNS-mined hard negatives while training UniDEC. Further, the table also shows the scores of inference done using either the DE head embedding $\Phi_{\mathfrak{D}}(x)$ or the normalized CLF head embedding $\mathfrak{N}(\Phi_{\mathfrak{C}}(x))$, instead of the concatenated vector $\{\Phi_{\mathfrak{D}}(x) \oplus \mathfrak{N}(\Phi_{\mathfrak{C}}(x))\}$. The P vs PSP trade-off associated with adding ANNS-mined hard-negatives is clear by observing the underlined values.

short-text loss over long-text document helps the model understand the label distribution better, this has a reverse effect on short-text datasets and the label distribution confuses with already short-text query distribution and ends up learning the label distribution more at the expense of query distribution.

## 5 Other Related Works

To reduce computational costs of training classifiers, previous XMC methods tend to make use of various shortlisting strategies, which serves as a good approximation to the loss over the entire label space [4, 7, 37]. This shortlist can be created in one of the two ways : (i) by training a meta classifier on coarser levels of a hierarchically-split probabilistic label tree. The leaf nodes of the top-k nodes constitute the shortlist [15, 18, 19] (ii) by retrieving the top-k labels for a query from an ANNS built on the label representations from a contrastively trained DE [5]. Both these methods have different trade-offs. The meta-classifier based approach has a higher memory footprint due to the presence of additional meta classifier ($\sim \mathbb{R}^{L/10 \times d}$ in size) along with the extreme classifier, but it gives enhanced performance since this provides progressively harder negatives in a dynamic shortlist, varying every epoch [15, 18, 19]. The shortlisting based on ANNS requires training the model in multiple stages, which has low memory usage, but needs longer training schedules and uses a static shortlist for training extreme classifiers [6, 7, 23, 24].

Previous research has also explored various other methods : (i) label trees [14, 17, 25, 34], (ii) classifiers based on hierarchical label

trees [4, 26, 38]. Tangentially, various initialisation methods [9, 31] and data augmentation approaches [20] have also been studied. Alongside previous negative-mining works, the statistical consequences of this sampling [30] and missing labels [12, 27, 32, 33, 35] have led to novel insights in designing unbiased loss functions - which can also be applied in UniDEC.

## 6 Conclusion

In this paper, we present a new loss-independent end-to-end XMC framework, UniDEC, that aims to leverage the best of both, a dual encoder and a classifier in a compute-efficient manner. The dual-encoder is used to mine hard negatives, which are in turn used as the shortlist for the classifier, eliminating the need for meta classifiers. Highly informative in-batch labels are created which maximise the supervisory signals while keeping the GPU memory footprint as low as possible - to the extent that we outperform previous SOTAs with just a single GPU. The dual encoders and classifiers are unified and trained with the same multi-class loss function, which follows the proposed pick-some-labels paradigm. To the best of our knowledge, we are the first work to study the effect of PAL-like losses for training XMC classifiers. We hope this inspires future works to study the proposed PSL reduction for multilabel problems as a compute-efficient means to further eliminate the need of high-capacity classifiers in XMC, bringing the scope of this problem closer to the more general dense retrieval regime.

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

# A Offline Evaluation and Live A/B testing on Sponsored Search

To demonstrate the effectiveness of our method on proprietary datasets and real-world scenarios, we do experiments in sponsored search setting. The proposed model was evaluated on proprietary dataset for matching queries to advertiser bid phrases (Query2Bid) consisting of 450M labels. Query2Bid-450M dataset was created by mining the logs from search engine and enhancing it through Data Augmentation techniques using a ensemble of leading (proprietary) algorithms such as Information Retrieval models (IR), Dense retrieval models (DR), Generative Non-Autoregressive models (NAR), Extreme-Multi-label Classification models (XMC) and even GPT Inference techniques.

*Experimental Setup :* The BERT Encoder is initialized with 6-Layer DistilBERT base architecture. Since the search queries and bid phrases are of short-text in nature, a max-sequence-length of 12 is used. We evaluate DePSL against XMC and DR models deployed in production which could scale to the magnitude of chosen dataset. Training batch-size is set to 2048 and other Hyperparameters are chosen to be same as for public benchmark datasets. Training is carried out on **8 V100 GPUs** and could easily complete within **48 hours**. Performance is measured using popular metrics such as Precision@K (P@K) with $K \in 1, 3, 5, 10$.

| Method | P@1 | P@3 | P@5 | P@10 |
|---|---|---|---|---|
| NGAME | 86.16 | 73.07 | 64.61 | 51.94 |
| SimCSE | 86.08 | 73.26 | 65.27 | 53.51 |
| DePSL | **87.33** | **74.63** | **66.44** | **54.13** |

**Table 6: Results on Query2Bid-450M dataset for Sponsored Search**

*Offline Results :* Table 6 shows that on DePSL can be 1.15-1.83% more accurate than the leading DR & XMC methods in Sponsored Search setting. This indicates that leveraging DePSL can yield superior gains in real-world search applications.

*Live A/B Testing in a Search Engine:* DePSL was deployed on Live Search Engine and A/B tests were performed on real-world traffic. The effect of adding DePSL to the ensemble of existing models in the system was measured through popular metrics such as Impression Yield (IY), Click Yield (CY), Click-Through Rate (CTR) and Query Coverage (QC). Refer [6] for definitions and details about these metrics. DePSL was observed to improve IY, CY, CTR and QC by **0.87%, 0.66%, 0.21% and 1.11%** respectively. Gains in IY, CY and CTR establish that DePSL is able to predict previously unmatched relations and the predictions are more relevant to the end user. QC boost indicates that DePSL is able to serve matches for queries to which there were no matches before in the system. This ascertains the zero-shot capabilities of the model.

# B Dataset Statistics

MS-MARCO, a representative dataset for DR tasks, has 3.2M documents but on average contains only 1.1 positively annotated answers (label) per question (instance) [28]. On the other hand, LF-AmazonTitles-1.3M, an XMC dataset which is representative dataset for product recommendation task, has a label space spanning 1.3M Amazon products where each instance (a product title) is annotated (tagged), by $\sim$ 22.2 labels (related product titles) and each label annotates, $\sim$ 38.2 instances. This indicates the broader spectrum of XMC tasks in contrast with zero-shot nature of ODQA task.

| Datasets | Benchmark | N | L | APpL | ALpP | AWpP |
|----------|-----------|---|---|------|------|------|
| MS-MARCO | DR | 502,931 | 8,841,823 | - | 1.1 | 56.58 |
| LF-AmazonTitles-131K | XMC | 294,805 | 131,073 | 5.15 | 2.29 | 6.92 |
| LF-Amazon-131K | XMC | 294,805 | 131,073 | 5.15 | 2.29 | 6.92 |
| LF-AmazonTitles-1.3M | XMC | 2,248,619 | 1,305,265 | 38.24 | **22.20** | 8.74 |
| LF-WikiSeeAlso-320K | XMC | 693,082 | 312,330 | 4.67 | 2.11 | 3.01 |
| Query2Bid-450M | Search Engine | 52,029,024 | 454,608,650 | 34.61 | 3.96 | - |

Table 7: Details of the benchmark datasets with label features. APpL stands for avg. points per label, ALpP stands for avg. labels per point and AWpP is the length i.e. avg. words per point.

| | P@1 | P@3 | P@5 | PSP@1 | PSP@3 | PSP@5 |
|---|-----|-----|-----|-------|-------|-------|
| Method | | | LF-WikiSeeAlsoTitles-320K | | | |
| UɴɪDEC | **36.3** | **23.2** | **17.3** | **26.3** | 27.8 | 29.9 |
| OAK | 33.7 | 22.7 | 17.1 | 25.8 | **28.5** | **30.8** |
| GraphSage | 27.3 | 17.2 | 13.0 | 21.6 | 21.8 | 23.5 |
| GraphFormer | 21.9 | 15.1 | 11.8 | 19.2 | 20.6 | 22.7 |
| NGAME | 32.6 | 22.0 | 16.6 | 24.4 | 27.4 | 29.9 |
| DEXA | 31.7 | 21.0 | 15.8 | 24.4 | 26.5 | 28.6 |
| ELIAS | 23.4 | 15.6 | 11.8 | 13.5 | 15.9 | 17.7 |
| CascadeXML | 23.4 | 15.7 | 12.1 | 12.7 | 15.4 | 17.6 |
| XR-Transformer | 19.4 | 12.2 | 9.0 | 10.6 | 11.8 | 12.7 |
| AttentionXML | 17.6 | 11.3 | 8.5 | 9.4 | 10.6 | 11.7 |
| SiameseXML | 32.0 | 21.4 | 16.2 | 26.8 | 28.4 | 30.4 |
| ECLARE | 29.3 | 19.8 | 15.0 | 22.0 | 24.2 | 26.3 |
| | | | LF-WikiTitles-500K | | | |
| UɴɪDEC | **50.2** | **28.8** | **23.3** | **25.9** | 25.2 | 24.9 |
| OAK | 44.8 | 25.9 | 17.9 | 25.7 | **25.8** | **25.0** |
| GraphSage | 27.2 | 15.7 | 11.3 | 22.3 | 19.3 | 19.1 |
| GraphFormer | 24.5 | 14.9 | 11.3 | 22.0 | 19.2 | 19.5 |
| NGAME | 39.0 | 23.1 | 16.1 | 23.1 | 23.3 | 23.0 |
| CascadeXML | 47.3 | 26.8 | 19.0 | 19.2 | 19.5 | 19.7 |
| AttentionXML | 40.9 | 21.5 | 15.0 | 14.8 | 14.0 | 13.9 |
| ECLARE | 44.4 | 24.3 | 16.9 | 21.6 | 20.4 | 19.8 |
| | | | LF-AmazonTitles-1.3M | | | |
| UɴɪDEC | **57.4** | **50.8** | **45.9** | **30.1** | **34.3** | **36.8** |
| GraphSage | 28.1 | 21.4 | 17.6 | 24.5 | 24.2 | 23.7 |
| GraphFormer | 24.2 | 17.4 | 14.3 | 22.5 | 22.4 | 22.5 |
| NGAME | 54.7 | 47.8 | 42.8 | 28.2 | 32.3 | 34.5 |
| DEXA | 56.6 | 49.0 | 43.9 | 29.1 | 32.7 | 34.9 |
| CascadeXML | 47.8 | 42.0 | 38.3 | 17.2 | 21.7 | 24.8 |
| XR-Transformer | 50.1 | 44.1 | 40.0 | 20.1 | 24.8 | 27.8 |
| PINA | 55.8 | 48.7 | 43.9 | - | - | - |
| AttentionXML | 45.0 | 39.7 | 36.2 | 16.0 | 19.9 | 22.5 |
| SiameseXML | 49.0 | 42.7 | 38.5 | 27.1 | 30.4 | 32.5 |
| ECLARE | 50.1 | 44.1 | 40.0 | 23.4 | 27.9 | 30.6 |

Table 9: Additional Results on short-text datasets

## C  Complete Results

| | P@1 | P@3 | P@5 | PSP@1 | PSP@3 | PSP@5 |
|---|-----|-----|-----|-------|-------|-------|
| **Method** | | | LF-WikiSeeAlso-320K | | | |
| UɴɪDEC | 47.69 | 30.74 | 22.81 | 35.45 | 38.02 | 40.71 |
| NGAME | 46.4 | 25.95 | 18.05 | 28.18 | 30.99 | 33.33 |
| DEXA | 47.11 | 30.48 | 22.71 | 31.81 | 35.5 | 38.78 |
| CascadeXML | 40.42 | 26.55 | 20.2 | 22.26 | 27.11 | 31.1 |
| XR-Transformer | 42.57 | 28.24 | 21.3 | 25.18 | 30.13 | 33.79 |
| PINA | 44.54 | 30.11 | 22.92 | - | - | - |
| AttentionXML | 40.5 | 26.43 | 19.87 | 22.67 | 26.66 | 29.83 |
| LightXML | 34.5 | 22.31 | 16.83 | 17.85 | 21.26 | 24.16 |
| SiameseXML | 42.16 | 28.14 | 21.39 | 29.02 | 32.68 | 36.03 |
| ECLARE | 40.58 | 26.86 | 20.14 | 26.04 | 30.09 | 33.01 |
| DECAF | 41.36 | 28.04 | 21.38 | 25.72 | 30.93 | 34.89 |
| Parabel | 33.46 | 22.03 | 16.61 | 17.1 | 20.73 | 23.53 |
| Bonsai | 34.86 | 23.21 | 17.66 | 18.19 | 22.35 | 25.66 |
| | | | LF-Wikipedia-500K | | | |
| UɴɪDEC | 83.8 | 62.63 | 47.17 | 42.11 | 49.32 | 51.78 |
| NGAME | 84.01 | 64.69 | 49.97 | 41.25 | 52.57 | 57.04 |
| DEXA | **84.92** | **65.5** | **50.51** | **42.59** | **53.93** | **58.33** |
| ELIAS | 81.26 | 62.51 | 48.82 | 35.02 | 45.94 | 51.13 |
| CascadeXML | 80.69 | 60.39 | 46.25 | 31.87 | 40.86 | 44.89 |
| XR-Transformer | 81.62 | 61.38 | 47.85 | 33.58 | 42.97 | 47.81 |
| PINA | 82.83 | 63.14 | 50.11 | - | - | - |
| AttentionXML | 82.73 | 63.75 | 50.41 | 34 | 44.32 | 50.15 |
| LightXML | 81.59 | 61.78 | 47.64 | 31.99 | 42 | 46.53 |
| SiameseXML | 67.26 | 44.82 | 33.73 | 33.95 | 35.46 | 37.07 |
| ECLARE | 68.04 | 46.44 | 35.74 | 31.02 | 35.39 | 38.29 |
| Parabel | 68.7 | 49.57 | 38.64 | 26.88 | 31.96 | 35.26 |
| Bonsai | 69.2 | 49.8 | 38.8 | - | - | - |

Table 8: Additional results on long text datasets

