# OpenReview forum: "UNIDEC : Unified Dual Encoder and Classifier Training for Extreme Multi-Label Classification"
_ACM.org/TheWebConf/2025/Conference — WWW 2025 Poster_

### Official Review · Reviewer_BDjw · 2024-11-29

**Novelty:** 5
**Technical Quality:** 5

**Review:**

The work presents a novel approach to addressing the Extreme Multi-Label Classification (XMC) task, demonstrating strong originality by unifying the training of a Dual Encoder and Classifier into a single framework. While the method is innovative, the extensive use of mathematical symbols and formulas can make it challenging to follow at times. Nonetheless, the work’s significance is evident, as it tackles critical challenges in XMC, such as computational inefficiency and scalability, contributing meaningfully to the field.

Strength
1. Achieves state-of-the-art results with up to 16× computational savings, making XMC training more accessible to resource-constrained settings.
2. Benchmarked across six diverse datasets (covering both short and long text), offering a comprehensive evaluation of the proposed approach.
3. Provides detailed insights through ablation studies, highlighting the contributions of individual components.

Weakness
1. The methodology section contains a large number of mathematical symbols, which can be difficult to follow and may hinder readability.
2. The presentation could be improved. For example, the figures should be vector graphics.

**Questions:**

1. How would UniDEC adapt if the label space grows beyond the current tested sizes (e.g.,
billions of labels)?
2. Could you clarify the configurations required to replicate results?

**Reviewer Confidence:**

3: The reviewer is confident but not certain that the evaluation is correct

**Scope:**

3: The work is somewhat relevant to the Web and to the track, and is of narrow interest to a sub-community

---

### Official Review · Reviewer_eAi9 · 2024-12-02

**Novelty:** 4
**Technical Quality:** 3

**Review:**

The paper proposes UniDEC, a framework for extreme multi-label classification (XMC). It unifies dual encoder and classifier training, leveraging a novel "pick-some-labels" (PSL) reduction to compute loss over subsets of labels. This approach reduces computational costs and memory requirements, enabling training on large datasets with millions of labels using a single GPU.
UniDEC achieves state-of-the-art performance across several public XMC datasets, outperforming existing dual encoder and classifier-based methods. It also demonstrates significant efficiency improvements, reducing GPU requirements by 4–16× compared to baseline models. These results establish UniDEC as both computationally efficient and effective for real-world XMC tasks.

The paper is well-written and structured, with clear descriptions of the proposed methodology, experimental setup, and results. However, some technical details, such as the ablation studies, might benefit from more concise explanations to improve accessibility for a broader audience.
The paper appropriately cites a wide range of related work, covering prior advancements in dual encoder methods, classifier-based approaches, and loss function optimizations for XMC.

There are a few typos in the text that could be fixed:
"upto" -> "up to"
"fit to Zipf’s law" -> "follows Zipf’s law"
"an inordinately large label pool of size" -> "a disproportionately large label pool of size"
In Figure 1: The phrase "denoting the the classifiers" has a duplicated "the."

Furthermore:
Dismec [2]" is mentioned, but the reference lacks complete details in the main text for first-time readers

The paper is challenging to follow for readers unfamiliar with extreme multi-label classification due to its reliance on specialized terminology and limited introductory explanations of key concepts.

**Questions:**

How does the random sampling of positive labels in the PSL reduction affect model performance on rare (tail) labels? Could this introduce instability in certain datasets?
The paper mentions that UniDEC is loss-independent. Have you tested other loss functions beyond SupCon and Decoupled Softmax, and if so, how do they perform?
Does the framework provide any interpretability into why certain labels are selected or ranked?

**Reviewer Confidence:**

1: The reviewer's evaluation is an educated guess

**Scope:**

1: The work is irrelevant to the Web

---

### Official Review · Reviewer_z536 · 2024-12-02

**Novelty:** 5
**Technical Quality:** 5

**Review:**

The paper proposes UniDEC, a unified framework combining dual encoder (DE) and classifier training for Extreme Multi-Label Classification (XMC). The approach introduces Pick-Some-Labels (PSL) reduction to efficiently calculate losses over a subset of labels, achieving state-of-the-art performance with significantly reduced computational costs. UniDEC shows notable improvements, particularly on datasets with a large number of labels, while maintaining computational efficiency, demonstrating its effectiveness.

The strength is that the methodology is solid, and the experiments are comprehensive, spanning a wide range of datasets and metrics. The weakness is that the clustering-based batching strategy, while effective, may introduce dependency on dataset-specific characteristics, which is not thoroughly analyzed.

**Questions:**

1. How do you determine the optimal number of positive and negative labels per instance in the PSL reduction? Have you explored dynamic tuning of these parameters during training?
2. What are the primary challenges faced when clustering queries for batching, especially for large-scale datasets? How sensitive is the clustering approach to dataset characteristics?
3. How does UniDEC scale beyond datasets like LF-AmazonTitles-1.3M or Query2Bid-450M? Would the current architecture and methods remain effective for datasets with billions of labels?
4. Have you explored the integration of external knowledge sources or pre-trained LLMs into the UniDEC framework?

**Reviewer Confidence:**

2: The reviewer is willing to defend the evaluation, but it is likely that the reviewer did not understand parts of the paper

**Scope:**

4: The work is relevant to the Web and to the track, and is of broad interest to the community

---

### Official Review · Reviewer_W8ag · 2024-12-06

**Novelty:** 4
**Technical Quality:** 5

**Review:**

This paper addresses challenges in XMC, a task of predicting a subset of relevant labels from a large label space (often in the millions), by proposing a framework named UNIDEC. It integrates dual encoder and classifier training into a single end-to-end trainable framework. The framework uses PSL to scale to datasets with millions of labels while reducing computational overhead. The experimental results on six datasets and ablation studies reveal the efficacy of the proposed method.

Pros:
1. The task is interesting and has real-world applications.
2. The proposed method is well-motivated.
3. The experimental results have thorough comparisons to baselines.
4. The code will be open-sourced upon acceptance.

Cons:
1. The writing of the paper can be improved. For example, the label for each equation is missing, the mathematical notation is quite heavy.
2. Limited discussion on the model’s computational overhead.
3. The symmetric loss formulation is a novel addition, but the paper could discuss its theoretical limitations or potential edge cases where it may lead to suboptimal results.
4. The dynamic batching and ANNS-based negative sampling introduce additional engineering complexity compared to simpler approaches.
5. The topic of the paper appears to align less closely with WWW. It is uncertain whether the WWW audience, typically interested in web-related technologies and large-scale systems, would find this topic compelling.

**Questions:**

Cons (1-5)

**Reviewer Confidence:**

2: The reviewer is willing to defend the evaluation, but it is likely that the reviewer did not understand parts of the paper

**Scope:**

3: The work is somewhat relevant to the Web and to the track, and is of narrow interest to a sub-community